# OFFLINE RL WITH OBSERVATION HISTORIES: ANALYZING AND IMPROVING SAMPLE COMPLEXITY

**Joey Hong**    **Anca Dragan**    **Sergey Levine**
UC Berkeley
{joey_hong,anca,sergey.levine}@berkeley.edu

## ABSTRACT

Offline reinforcement learning (RL) can in principle synthesize more optimal behavior from a dataset consisting only of suboptimal trials. One way that this can happen is by "stitching" together the best parts of otherwise suboptimal trajectories that overlap on similar states, to create new behaviors where each individual state is in-distribution, but the overall returns are higher. However, in many interesting and complex applications, such as autonomous navigation and dialogue systems, the state is partially observed. Even worse, the state representation is unknown or not easy to define. In such cases, policies and value functions are often conditioned on *observation histories* instead of states. In these cases, it is not clear if the same kind of "stitching" is feasible at the level of observation histories, since two different trajectories would always have different histories, and thus "similar states" that might lead to effective stitching cannot be leveraged. Theoretically, we show that standard offline RL algorithms conditioned on observation histories suffer from poor sample complexity, in accordance with the above intuition. We then identify sufficient conditions under which offline RL can still be efficient – intuitively, it needs to learn a compact representation of history comprising only features relevant for action selection. We introduce a *bisimulation loss* that captures the extent to which this happens, and propose that offline RL can explicitly optimize this loss to aid worst-case sample complexity. Empirically, we show that across a variety of tasks either our proposed loss improves performance, or the value of this loss is already minimized as a consequence of standard offline RL, indicating that it correlates well with good performance.

## 1 INTRODUCTION

Deep reinforcement learning (RL) has achieved impressive performance in games (Mnih et al., 2013; Silver et al., 2017; AlphaStar, 2019), robotic locomotion (Schulman et al., 2015; 2017), and control (Todorov et al., 2012; Haarnoja et al., 2018). A key challenge in the widespread adoption of RL algorithms is the need for deploying a suboptimal policy in the environment to collect online interactions, which can be detrimental in many applications such as recommender systems (Afsar et al., 2021), healthcare (Shortreed et al., 2011; Wang et al., 2018), and robotics (Kalashnikov et al., 2018). Offline RL aims to learn effective policies entirely from an offline dataset of previously collected demonstrations (Levine et al., 2020), which makes it a promising approach for applications where exploring online from scratch is unsafe or costly. A major reason for the success of offline RL algorithms is their ability to combine components of suboptimal trajectories in the offline dataset using common states, a phenomenon called "trajectory stitching" (Fu et al., 2019a; 2020).

Most offline RL methods are formulated in a Markov decision process (MDP) where the state is fully observed (Sutton and Barto, 2018). However, in many real-world tasks, the state is only partially observed, corresponding to a partially observable Markov decision process (POMDP) (Spaan). For example, in autonomous driving, the robot is limited to information measured by sensors, and does not directly perceive the positions of every car on the road, much less the intentions of every driver. As another example, in dialogue systems, the conversational agent can only observe (potentially noisy and redundant) utterances of the other agents, while their underlying preferences and mental state are hidden. In fact, there is often not even a clear representation or parameterization of "state" (e.g., what is the space of human intentions or preferences?). Therefore, in such applications, policies must

instead be conditioned on all observations thus far – the *observation history*. Naïvely, this leads to concerns on the efficiency of existing offline RL algorithms. Offline RL is much less likely to utilize suboptimal behaviors if stitching them requires shared observation histories among them, as histories are much less likely to repeat in datasets that are not prohibitively large.

In this work, we aim to answer the following question: *When and how can we improve the sample efficiency of offline RL algorithms when policies are conditioned on entire observation histories?* Given that observation histories make naïve stitching very inefficient, we study this question from the lens of when and how we can enable history-conditioned offline RL to efficiently leverage trajectory stitching. Our focus is on a theoretic analysis of this question, though we also provide simple empirical evaluations to confirm our findings. Theoretically, we first show that in the worst case, naïve offline RL using observation histories can lead to very poor sample complexity bounds. We show that prior pessimistic offline RL algorithms with near-optimal sample complexity guarantees in fully observed MDPs (Rashidinejad et al., 2021; Jin et al., 2021a) fail to learn efficiently with observation histories. We also propose a remedy to this, by learning a compact representation of histories that contains only the relevant information for action selection. When these representations induce a *bisimulation metric* over the POMDP, we prove that offline RL algorithms achieve greatly improved sample complexity. Furthermore, when existing offline RL algorithms fail to learn such representations, we propose a novel modification that explicitly does so, by optimizing an auxiliary *bisimulation loss* on top of standard offline RL objective. Empirically, we show – in simple navigation and language model tasks – that when naïve offline RL algorithms fail, using our proposed loss in conjunction with these algorithms improves performance; furthermore, we also show that in tasks where existing offline RL approaches already succeed, our loss is implicitly being minimized. Our work provides, to our knowledge, the first theoretical treatment of representation learning in partially observed offline RL, and offers a step toward provably efficient RL in such settings.

## 2 RELATED WORK

**Offline RL.** Offline RL (Lange et al., 2012; Levine et al., 2020) has shown promise in a range of domains. To handle distribution shift (Fujimoto et al., 2018; Kumar et al., 2019), many modern offline RL algorithms adopt a pessimistic formulation, learning a lower-bound estimate of the value function or Q-function (Kumar et al., 2020; Kostrikov et al., 2021; Kidambi et al., 2020; Yu et al., 2020; 2021). When they work properly, offline RL algorithms should benefit from "trajectory stitching," or combining components of suboptimal trajectories in the data to make more optimal ones (Fu et al., 2019a; 2020). From a theoretical perspective, multiple prior works show that pessimistic offline RL algorithms have near-optimal sample complexity, under assumptions on the affinity between the optimal and behavior policies (Liu et al., 2020; Rashidinejad et al., 2021; Xie et al., 2021; Jin et al., 2021b). Notably, Xie et al. (2021) show that pessimistic offline RL algorithms can attain the information-theoretic lower-bound in tabular MDPs, and Jin et al. (2021b) show a similar result for linear MDPs. In our work, we consider offline RL where policies condition on observation histories.

**POMDPs.** Our work studies offline RL in POMDPs. A number of prior works on RL in POMDPs have proposed designing models, such as RNNs, that can process observation histories (Zhang et al., 2015; Heess et al., 2015). Other methods instead aim to learn a model of the environment, for example via spectral methods (Azizzadenesheli et al., 2016) or Bayesian approaches that maintains a belief state over the environment parameters (Ross et al., 2011; Katt et al., 2018). However, such approaches can struggle to scale to large state and observation spaces. Igl et al. (2018) propose approximately learning the belief state using variational inference, which scales to high-dimensional domains but does not have any theoretical guarantees. To our knowledge, provably efficient offline RL methods for POMDPs are still relatively sparse in the literature. Recently, Jin et al. (2020) propose estimating the parameters of a tabular POMDP efficiently using the induced observable operator model (Jaeger, 2000), under an undercompleteness assumption between the observations and hidden state. Guo et al. (2022) propose and analyze a similar approach for linear POMDPs. However, these approaches share the same weaknesses as prior methods that rely on spectral methods in that they do not scale beyond linear domains. In our work, we analyze practical offline RL algorithms that work on general POMDPs, and show sufficient conditions on how they can be provably efficient, as well as propose a new algorithm that satisfies these conditions.

**Representation learning in RL.** Motivated by our theoretical analysis of the efficiency of naïve history-based policies, we propose an approach for learning compact representations of observations histories to improve the efficiency of offline RL in POMDPs. Multiple prior works consider state

abstraction in MDPs, often by learning low-dimensional representations using reconstruction (Hafner et al., 2019; Watter et al., 2015) or a contrastive loss (van den Oord et al., 2018). Specifically, our work builds on *bisimulation metrics* (Ferns et al., 2012; Castro, 2019), which identify equivalence classes over states based on rewards and transition probabilities. Recently, Zhang et al. (2021) propose learning representations that follow bisimulation-derived state aggregation to improve deep RL algorithms, and Kemertas and Aumentado-Armstrong (2021) propose extensions that improve robustness. The main objective of our work is not to propose a new representation learning algorithm, but to identify when offline RL with observation histories can achieve efficient sample complexity in POMDPs. To our knowledge, we are the first to provably show efficient offline RL in POMDPs using theoretical guarantees derived from representation learning.

## 3 PRELIMINARIES

The goal in our problem setting is to learn a policy $\pi$ that maximizes the expected cumulative reward in a partially observable Markov decision process (POMDP), given by a tuple $M = (\mathcal{S}, \mathcal{A}, \mathcal{O}, \mathcal{T}, r, \mathcal{E}, \mu_1, H)$, where $\mathcal{S}$ is the state space, $\mathcal{A}$ is the action space, $\mathcal{O}$ is the observation space, $\mu_1$ is the initial state distribution, and $H$ is the horizon. When action $a \in \mathcal{A}$ is executed at state $s \in \mathcal{S}$, the next state is generated according to $s' \sim \mathcal{T}(\cdot|s, a)$, and the agent receives stochastic reward with mean $r(s, a) \in [0, 1]$. Subsequently, the agent receives an observation $o' \sim \mathcal{E}(\cdot|s')$.

Typically, POMDPs are defined with a state space representation; in practice though, these are notoriously difficult to define, and so instead we transform POMDPs into MDPs over observation histories – henceforth called *observation-history-MDPs* (Timmer, 2010). At timestep $h \in [H]$, we define the *observation history* $\tau_h$ as the sequence of observations and actions $\tau_h = [o_1, a_1, o_2, \dots, o_h]$. Then, an observation-history-MDP is given by tuple $M = (\mathcal{H}, \mathcal{A}, P, r, \rho_1, H)$, where $\mathcal{H}$ is the space of observation histories, and $\mathcal{A}$ is the action space, $\rho_1$ is the initial observation distribution, and $H$ is the horizon. When action $a \in \mathcal{A}$ is executed at $h \in \mathcal{H}$, the agent observes $h' = h \oplus o'$ via $o' \sim P(\cdot|\tau, a)$, where $\oplus$ denotes concatenation, and receives reward with mean $r(\tau, a)$.

The Q-function $Q^\pi(\tau, a)$ for a policy $\pi(\cdot|\tau)$ represents the discounted long-term reward attained by executing $a$ given observation history $\tau$ and then following policy $\pi$ thereafter. $Q^\pi$ satisfies the Bellman recurrence: $Q^\pi(\tau, a) = \mathbb{B}^\pi Q^\pi(\tau, a) = r(\tau, a) + \mathbb{E}_{h' \sim \mathcal{T}(\cdot|\tau, a), a' \sim \pi(\cdot|\tau')}[Q_{h+1}(\tau', a')]$. The value function $V^\pi$ considers the expectation of the Q-function over the policy $V^\pi(h) = \mathbb{E}_{a \sim \pi(\cdot|\tau)}[Q^\pi(\tau, a)]$. Meanwhile, the Q-function of the optimal policy $Q^*$ satisfies: $Q^*(\tau, a) = r(\tau, a) + \mathbb{E}_{h' \sim \mathcal{T}(\cdot|\tau, a)}[\max_{a'} Q^*(\tau', a')]$, and the optimal value function is $V^*(\tau) = \max_a Q^*(\tau, a)$. Finally, the expected cumulative reward is given by $J(\pi) = \mathbb{E}_{o_1 \sim \rho_1}[V^\pi(\tau_1)]$. Note that we do not condition the Q-values nor policy on timestep $h$ because it is implicit in the length of $\tau$.

In offline RL, we are provided with a dataset $\mathcal{D} = \{(\tau_i, a_i, r_i, o'_i)\}_{i=1}^N$ of size $|\mathcal{D}| = N$. We assume that the dataset $\mathcal{D}$ is generated i.i.d. from a distribution $\mu(\tau, a)$ that specifies the effective behavior policy $\pi_\beta(a|\tau) := \mu(\tau, a)/\sum_a \mu(\tau, a)$. In our analysis, we will use $n(\tau, a)$ to denote the number of times $(\tau, a)$ appears in $\mathcal{D}$, and $\widehat{P}(\cdot|\tau, a)$ and $\widehat{r}(\tau, a)$ to denote the empirical dynamics and reward distributions in $\mathcal{D}$, which may be different from $P$ and $r$ due to stochasticity and sampling error. Finally, as in prior work (Rashidinejad et al., 2021; Kumar et al., 2022), we define the suboptimality of learned policy $\widehat{\pi}$ as $\mathsf{SubOpt}(\widehat{\pi}) = \mathbb{E}_{\mathcal{D} \sim \mu}[J(\pi^*) - J(\widehat{\pi})]$.

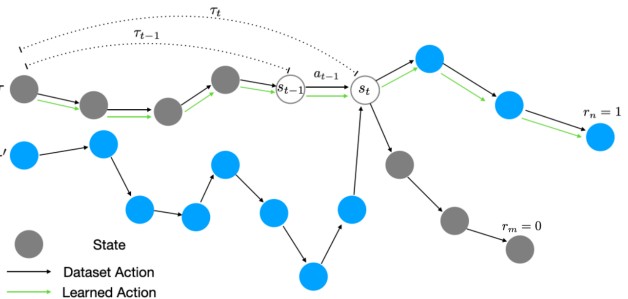

Figure 1: Illustrative example of trajectory stitching. Here, Q-learning is able to learn that though the grey trajectory $\tau$ was unsuccessful, a prefix $\tau_t$ of the trajectory is still optimal when stitched with the suffix of blue trajectory $\tau'$.

**Trajectory stitching.** Much of how offline RL can learn efficiently lies in its capability to combine components of suboptimal trajectories to deduce better ones, which is called "trajectory stitching". We illustrate this in Figure 1, where a trajectory $\tau$ through state $s_{t-1}$ does not end in positive reward, but does share a common state $s_t$ with trajectory $\tau'$ that does. In MDPs, offline RL using value iteration

will learn Q-values: $\widehat{Q}(s_{t-1}, a_{t-1}) = \sum_{s'} P(s'|s_{t-1}, a_{t-1})\widehat{V}(s')$. Because $\widehat{V}(s_t)$ is known to be positive from observing $\tau'$, offline RL can deduce that taking action $a_{t-1}$ at $s_{t-1}$ also has positive value, without explicitly observing it in the dataset. This becomes complicated in an observation history MDP, as offline RL will now learn $\widehat{Q}(\tau_{t-1}, a_{t-1}) = \sum_{s'} P(s'|s_{t-1}, a_{t-1})\widehat{V}(\tau_t)$. But $\widehat{V}(\tau_t)$ is not known to be positive because $\tau_t$ has not been observed in the data. This means that, naïvely, offline RL on observation history MDPs does not seem to benefit from trajectory stitching, which may negatively effect how efficiently it can learn from data. We formalize this in Section 4 by proving that offline RL can have poor worst-case sample complexity in POMDPs.

**Notation.** Let $n \wedge 1 = \max\{n, 1\}$. Denote $\iota = \text{polylog}(|\mathcal{O}|, |\mathcal{A}|, H, N)$. We let $\iota$ be a polylogarithmic quantity, changing with context. For $d$-dimensional vectors $x, y$, $x(i)$ denotes its $i$-th entry, and define $\mathbb{V}(x, y) = \sum_i x(i)y(i)^2 - (\sum_i x(i)y(i))^2$.

# 4 SHOWING INEFFICIENCY OF OFFLINE RL IN OBSERVATION-HISTORY-MDPs

In this section, we show that existing offline RL algorithms with state-of-the-art sample complexity guarantees in standard MDPs have significantly worse guarantees in observation history MDPs. We consider a class of offline RL algorithms that learn pessimistic value functions such that the estimated value lower-bounds the true one, i.e., $\widehat{V}^\pi \leq V^\pi$ for policy $\pi$. Practical implementations achieve this by subtracting a penalty from the reward, either explicitly (Yu et al., 2020; Kidambi et al., 2020) or implicitly (Kumar et al., 2020; Kostrikov et al., 2021). We only analyze one such algorithm that does the former, though our findings can likely be extended to general pessimistic offline RL methods.

We consider a meta-algorithm called pessimistic value iteration (PEVI), originally introduced by Jin et al. (2021a). This algorithm relies on the construction of confidence intervals $c : \mathcal{H} \times \mathcal{A} \to \mathbb{R}$ that are high-probability bounds on the estimation error of $\widehat{P}, \widehat{r}$. Then, pessimistic Q-values are obtained by solving the Bellman recurrence: $\widehat{Q}(\tau, a) \leftarrow \widehat{r}(\tau, a) - c(\tau, a) + \sum_{o'} \widehat{P}(o'|\tau, a)\widehat{V}(\tau')$, where values are $\widehat{V}(\tau) \leftarrow \sum_a \widehat{Q}(\tau, a)\widehat{\pi}(a|\tau)$. The learned policy is then $\widehat{\pi}(\cdot|\tau) \leftarrow \arg\max_\pi \sum_a \widehat{Q}(\tau, a)\pi(a|\tau)$. We give a full pseudocode of the algorithm in Algorithm 2 in Appendix A.1.

Prior work has shown that in tabular MDPs, instantiations of PEVI achieve state-of-the-art sample complexity (Rashidinejad et al., 2021). We choose one such instantiation, where confidence intervals $c(\tau, a)$ are derived using Bernstein's inequality:

$$c(\tau, a) \leftarrow \sqrt{\frac{H\mathbb{V}(\widehat{P}(\cdot|\tau, a), \widehat{V}(\tau \oplus \cdot))\iota}{(n(\tau, a) \wedge 1)}} + \sqrt{\frac{H\widehat{r}(\tau, a)\iota}{(n(\tau, a) \wedge 1)}} + \frac{H\iota}{(n(\tau, a) \wedge 1)}. \tag{1}$$

The same instantiation was considered by Kumar et al. (2022), and shown to achieve sample complexity approaching the information-theoretic lower-bound. The additional dependence on $H$ is due to $\log|\mathcal{H}| = H\,\text{polylog}(|\mathcal{O}|, |\mathcal{A}|)$. However, we can show that in an observation history MDP, the same algorithm has much worse sample complexity bounds.

We first characterizes the distribution shift between the offline dataset distribution $\mu(\tau, a)$ and the distribution induced by $\pi^*$, given by $d^*(\tau, a)$, via a *concentrability coefficient* $C^*$.

**Definition 4.1** (Concentrability of the data distribution). *Define $C^*$ to be the smallest finite constant that satisfies $d^*(\tau, a)/\mu(\tau, a) \leq C^*$ $\forall \tau \in \mathcal{H}, a \in \mathcal{A}$.*

Intuitively, the coefficient $C^*$ formalizes how well the data distribution $\mu(\tau, a)$ covers the tuples $(\tau, a)$ visited under the optimal $\pi^*$, where $C^* = 1$ corresponds to data from $\pi^*$ and increases with distribution shift. $C^*$ was first introduced by Rashidinejad et al. (2021) but for standard MDPs. Using $C^*$, we can derive the following sample-complexity bound for PEVI in an observation history MDP:

**Theorem 4.1** (Suboptimality of PEVI in Tabular POMDPs). *In a tabular POMDP, the policy $\widehat{\pi}^*$ found by PEVI satisfies*

$$\text{SubOpt}(\widehat{\pi}) \lesssim \sqrt{\frac{C^*|\mathcal{H}|H^3\iota}{N}} + \frac{C^*|\mathcal{H}|H^2\iota}{N}.$$

We defer our proof, which follows from adapting existing analysis from standard MDPs to observation history MDPs, to Appendix A. Note that dependence on $|\mathcal{H}|$ makes the bound exponential in the horizon because the space of observation histories satisfies $|\mathcal{H}| > |\mathcal{O}|^H$. This term arises due to

encountering observation histories that do not appear in the dataset; without additional assumptions on the ability to generalize to unseen histories, any offline RL algorithm must incur this suboptimality (as it can only take actions randomly given such histories), making the above bound tight.

## 5   ANALYZING WHEN SAMPLE-EFFICIENCY CAN BE IMPROVED

In this section, we show how the efficiency of offline RL algorithms can be improved by learning *representations* of observation histories, containing features of the history that sufficiently capture what is necessary for action selection. We then provide one method for learning such representations based on *bisimulation metrics* that, when used alongside existing offline RL algorithms, is sufficient to greatly improve their sample complexity guarantees in observation-hisotry MDPs.

Intuitively, consider that observation histories likely contains mostly irrelevant or redundant information. This means that it is possible to learn *summarizations*, such that instead of solving the observation history MDP, it is sufficient to solve a *summarized MDP* where the states are summarizations, actions are unchanged, and the dynamics and reward function are parameterized by the summarizations rather than observation histories. We formalize our intuition into the following:

**Assumption 5.1.** *There exists a set $\mathcal{Z}$ where $|\mathcal{Z}| \ll |\mathcal{H}|$, and $\varepsilon > 0$, such that the summarized MDP $(\mathcal{Z}, \mathcal{A}, P, r, \rho_1, H)$ satisfies: for every $\tau \in \mathcal{H}$ there exists a $z \in \mathcal{Z}$ satisfying $|V^*(\tau) - V^*(z)| \le \varepsilon$.*

The implication of Assumption 5.1 is that we can abstract the space of observation histories into a much more compact space of summarizations, containing only features of the history relevant for action selection. If the state space was known, then summarizations could be constructed as beliefs over the true state. In our case, one practical way of creating summarizations is by aggregating observation histories using the distances between their learned representations. Note that these representations may be implicitly learned by optimizing the standard offline RL objective, or they can be explicitly learned via an auxiliary representation learning objective. We describe one possible objective in the following section, which enjoys strong theoretical guarantees.

### 5.1   ABSTRACTING OBSERVATION HISTORIES USING BISIMULATION METRICS

*Bisimulation metrics* offer one avenue for learning abstractions of the observation history (Ferns et al., 2012; Castro, 2019). While they are not the only way of learning useful representations, these metrics offer strong guarantees for improving the efficiency of learning in standard MDPs, and are also empirically shown to work well with popular off-policy RL algorithms (Zhang et al., 2021). In contrast, we leverage learning bisimulation metrics and show that they can similarly improve the theoretical and empirical performance of offline RL algorithms in observation-history MDPs.

Formally, we define the *on-policy bisimulation metric* for policy $\pi$ on an observation-history-MDP as
$$d^\pi(\tau, \tau') = |r^\pi(\tau) - r^\pi(\tau')| + W_1\left(P^\pi(\cdot \mid \tau), P^\pi(\cdot \mid \tau')\right) , \tag{2}$$
where we superscript the reward and transition function by $\pi$ to indicate taking an expectation over $\pi$. To simplify notation, let $d^* = d^{\pi^*}$ be shorthand for the $\pi^*$-*bisimulation metric*.

Rather than using the true bisimulation metric, Zhang et al. (2021) showed that it can be more practical to learn an approximation of it in the embedding space. Similarly, we propose learning an encoder $\phi : \mathcal{H} \to \mathbb{R}^d$ such that distances $\widehat{d}_\phi(\tau, \tau') = ||\phi(\tau) - \phi(\tau')||_2^2$ approximate the distance under the $\pi^*$-bisimulation metric $d^*(\tau, \tau')$. Such an encoder can be learned implicitly by minimizing the standard offline RL objective, or explicitly by via an auxilliary MSE objective: $\phi = \arg\min ||\widehat{d}_\phi - d^*||_2^2$.

Then, the encoder can be used to compact the space of observation histories $\mathcal{H}$ into a space of summarizations $\mathcal{Z}$ by introducing an *aggregator* $\Phi : \mathcal{H} \to \mathcal{Z}$ that maps observation histories to summarizations. Specifically, the aggregator will cluster observation histories that are predicted to be similar under our learned bisimulation metric, i.e., $\Phi(\tau) = \Phi(\tau')$ for $\tau, \tau' \in \mathcal{H}$ if $\widehat{d}_\phi(\tau, \tau') \le \varepsilon$. This means that we can approximate the current observation history MDP with a *summarized MDP* $(\mathcal{Z}, \mathcal{A}, P, r, \rho_1, H)$. Any practical offline RL algorithm can be used to solve for the policy $\widehat{\pi}$ on the summarized MDP, and the policy can be easily evaluated on the original POMDP by selecting actions according to $\widehat{\pi}(\cdot \mid \Phi(\tau))$. In the following section, we show that doing so yields greatly improved sampled complexity guarantees in the original POMDP.

### 5.2   THEORETICAL ANALYSIS

In Section 4, we showed that applying a naïve pessimistic offline RL algorithm (PEVI), which has optimal sample complexity in standard MDPs, to observation-history-MDPs can incur suboptimality

that scales very poorly (potentially exponentially) with horizon $H$. Here, we show that applying the same algorithm to a summarized MDP, which aggregates observation histories based on how similar their learned representations are, can achieve greatly improved sample-complexity guarantees in the original observation-history-MDP, if the representations induce a bisimulation metric.

The first result we show relates the value functions under the original observation-history-MDP and a summarized MDP induced via the summarization function $\Phi$:

**Lemma 5.1.** *Let* $\Phi : \mathcal{H} \to \mathcal{Z}$ *be a learned aggregator that clusters observation histories such that* $\Phi(\tau) = \Phi(\tau') \Rightarrow \widehat{d}_\phi(\tau, \tau') \leq \varepsilon$. *Then, the induced summarized MDP* $(\mathcal{Z}, \mathcal{A}, P, r, \rho_1, H)$ *satisfies*

$$|V^*(\tau) - V^*(\Phi(\tau))| \leq H \left( \varepsilon + \left\| \widehat{d}_\phi - d^* \right\|_\infty \right).$$

Next, we show an improved sample complexity bound than Theorem 4.1 in a tabular MDP. We consider the same instantiation of PEVI as in Section 4. However, rather than operating on the raw observation history $\tau$, we use the summarization function $\Phi(\tau)$ obtained by learning a bisimulation metric over the space of histories $\mathcal{H}$. We can show that operating on the space of summarizations $\mathcal{Z}$ instead of the observation histories $\mathcal{H}$ leads to the following greatly improved bound:

**Theorem 5.1** (Suboptimality of PEVI augmented with $\Phi$ in Tabular POMDPs). *In a tabular POMDP, the policy* $\widehat{\pi}$ *found by* PEVI *on the summarized MDP* $(\mathcal{Z}, \mathcal{A}, P, r, \rho_1, H)$ *satisfies*

$$\mathsf{SubOpt}(\widehat{\pi}) \lesssim \sqrt{\frac{C^* |\mathcal{Z}| H^3 \iota}{N}} + \frac{C^* |\mathcal{Z}| H^2 \iota}{N} + 2H \left( \varepsilon + \left\| \widehat{d}_\phi - d^* \right\|_\infty \right).$$

Again, we defer full proofs to Appendix A. Here, we see that rather than exponential scaling in horizon $H$, offline RL now enjoys near optimal scaling, particularly if $|\mathcal{Z}| \ll |\mathcal{H}|$.

## 6 PRACTICAL APPROACH TO IMPROVING OFFLINE RL ALGORITHMS

As described in Section 5, the key component that enables sample-efficient offline RL is the existence of an encoder $\phi : \mathcal{H} \to \mathbb{R}^d$ that learns compact representations of observation histories. Specifically, we showed that if the distances between representations under the encoder $\widehat{d}_\phi(\tau, \tau') = ||\phi(\tau) - \phi(\tau')||_2^2$ match the $\pi^*$-bisimulation metric, offline RL algorithms that leverage these representations enjoy better efficiency when required to condition on observation histories. Note that the bound in Theorem 4.1 is a worst-case result. In the general case, even naïve offline RL algorithms might still *naturally* learn encoders $\phi$ as part of the standard training process that produce useful representations. In Section 5, we show that one way of measuring the effectiveness of the representations is by how well they induce a bisimulation metric. Though this is not a *necessary* condition, we do show that it is *sufficient* for sample-efficient learning. Therefore, in this section, we propose a practical way to train an encoder $\phi$ to induce a bisimulation metric.

---

**Algorithm 1** Offline RL with Bisimulation Learning

**Require:** Offline dataset $\mathcal{D}$
1: Initialize encoders $\phi, \bar{\phi}$
2: **for** $i = 1, 2, \ldots$ **do**
3:     Train encoder $\phi$: $J(\phi)$
4:     Train dynamics $\widehat{r}_\phi, \widehat{P}_\phi$: $J(\widehat{r}, \phi), J(\widehat{P}, \phi)$
5:     Train policy $\widehat{\pi}_\phi$
6:     Update target encoder: $\bar{\phi} \leftarrow (1 - \alpha)\bar{\phi} + \alpha\phi$
7: Return $\widehat{\pi}_\phi$

---

Note that in practice, we cannot naïvely fit $\widehat{d}_\phi$ to the $\pi^*$-bisimulation metric $d^*$, because it assumes knowledge of: (1) the true reward function $r$ and observation dynamics $P$ of the environment, and (2) the optimal policy $\pi^*$. To remedy this, we propose a practical algorithm similar to the one proposed by Zhang et al. (2021), where an encoder $\phi$ and policy $\widehat{\pi}_\phi$, operating on the embeddings, are trained jointly. To resolve (1), we fit a reward and dynamics model $\widehat{r}_\phi, \widehat{P}_\phi$ using dataset $\mathcal{D}$. Then, to resolve (2), we use the learned policy $\widehat{\pi}_\phi$ rather than optimal $\pi^*$, which intuitively should converge to $\pi^*$.

Formally, given the current learned policy $\widehat{\pi}_\phi$ with encoder $\phi$, we train $\phi$ with the *bisimulation loss* on top of the regular offline RL objective, using the following loss function:

$$J(\phi) = \mathbb{E}_{\substack{\tau, \tau' \sim \mathcal{D}, a \sim \widehat{\pi}(\cdot|z) \\ a' \sim \widehat{\pi}(\cdot|z')}} \left[ \left( \|\phi(\tau) - \phi(\tau')\| - |\widehat{r}(z, a) - \widehat{r}(z', a')| - D(\widehat{P}(\cdot \mid z, a), \widehat{P}(\cdot \mid z', a')) \right)^2 \right],$$

where $z = \bar{\phi}(\tau), z' = \bar{\phi}(\tau')$ are the representations from a target network. We choose $D$ to be an approximation of the 1-Wasserstein distance; in discrete observation settings, we use total variation

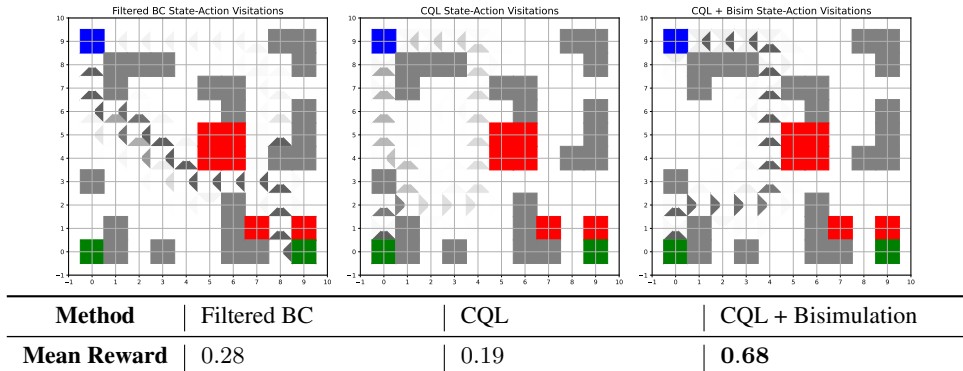

| Method | Filtered BC | CQL | CQL + Bisimulation |
|---|---|---|---|
| Mean Reward | 0.28 | 0.19 | **0.68** |

Figure 2: In our gridworld environment, Filtered BC takes the path towards the unsafe goal, CQL tries to take the path towards the safe goal but often incorrectly (by going down instead of right), and CQL with bisimulation loss always takes the correct path towards the safe goal.

$||\widehat{P}(\cdot|z, a) - \widehat{P}(\cdot|z', a')||_1$, and in continuous settings, we use $W_2(\widehat{P}(\cdot|z, a), \widehat{P}(\cdot|z', a'))$ on Gaussian distributions. Then, we use $\phi$ to train our dynamics model of the environment via the following:

$$J(\widehat{r}, \phi) = \mathbb{E}_{\tau,a,r\sim\mathcal{D}} \left[ (r - \widehat{r}(\phi(\tau), a))^2 \right] , \quad J(\widehat{P}, \phi) = \mathbb{E}_{\tau,a,\tau'\sim\mathcal{D}} \left[ d\left( z', \widehat{P}(\phi(\tau), a) \right) \right] ,$$

where again $z' = \bar{\phi}(\tau')$. Here, $d$ is either log-likelihood or mean-squared error if observations are discrete or continuous, respectively. Finally, we perform policy improvement on $\widehat{\pi}$, which conditions on representations generated by $\phi$. Note that this can be any existing offline RL algorithm. We detail pseudocode for the meta-algorithm in Algorithm 1.

## 7 EXPERIMENTS

Our experimental evaluation aims to empirically analyze the relationship between the performance of offline RL in partially observed settings and the bisimulation loss we discussed in Section 6. Our hypothesis is that, if naïve offline RL performs poorly on a given POMDP, then adding the bisimulation loss should improve performance, and if offline RL already does well, then the learned representations should **already** induce a bisimulation metric, and thus a low value of this loss. Note that our theory does not state that naïve offline RL will always perform poorly, just that it has a poor worst-case bound, so we would not expect an explicit bisimulation loss to always be necessary, though we hypothesize that successful offline RL runs might still minimize loss as a byproduct of successful learning when they work well. We describe the main elements of each evaluation in the main paper, and defer implementation details to Appendix B.

### 7.1 TABULAR NAVIGATION

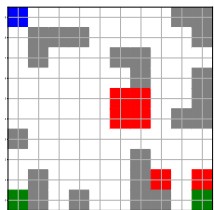 We first evaluate our hypothesis in a task involving navigation in a $10 \times 10$ tabular environment similar to gridworld (Fu et al., 2019b). Like gridworld, the environment we consider contains a start (blue) and goal (green) state, and walls (grey) and lava (red) placed in between. We consider a sparse reward where the agent earns a reward of 1 upon reaching the goal state; however, if the agent reaches a lava state, then its reward is 0 for the rest of the trajectory. The agent is able to move in either of the four directions (or choose to stay still). To introduce stochasticity in the transition dynamics, there is a 20% chance that the agent travels in a different direction (that is uniformly sampled) than commanded. Finally, the horizon of each episode is $H = 50$. Unlike conventional gridworld, the location of the goal state in our environment changes depending on what states the agent visits earlier in the trajectory. The specific layout is shown on the left. If the agent takes downwards path from the start state, they will trip a switch that turns the goal into the state in the lower right surrounded by lava; otherwise, the goal is the state in the lower left. Because the location of the goal state is unknown and depends on past behavior, it must be inferred from the observation history of the agent. Because the goal state in the lower left is "safe" (i.e. not surrounded by lava), an optimal agent should intentionally trip the switch by going right.

We construct a dataset of size $|\mathcal{D}| = 5,000$ where 50% of trajectories come from a policy that moves randomly, and 50% from a policy that primarily takes the path towards the "unsafe" goal state in

| Method | Mean Reward (Base Task) | Mean Reward (Hard Task) |
|---|---|---|
| BC | $0.05 \pm 0.02$ | $0.01 \pm 0.01$ |
| Filtered BC | $0.41 \pm 0.12$ | $0.12 \pm 0.05$ |
| CQL | $0.64 \pm 0.17$ | $0.43 \pm 0.08$ |
| CQL + Bisimulation | $\mathbf{0.71} \pm 0.14$ | $0.58 \pm 0.09$ |
| IQL | $0.63 \pm 0.15$ | $0.48 \pm 0.07$ |
| IQL + Bisimulation | $0.70 \pm 0.15$ | $\mathbf{0.61} \pm 0.11$ |
| MOPO | $0.55 \pm 0.18$ | $0.41 \pm 0.12$ |

Table 1: Mean and standard deviation of scores achieved on ViZDoom navigation task.

the lower right. We train three algorithms on this dataset, all of which use an RNN to process the observation histories: (1) filtered behavior cloning (BC) on the $25\%$ of trajectories in the data with highest reward, (2) conservative Q-learning (CQL) (Kumar et al., 2020), which is a strong offline RL baseline, and (3) CQL augmented with our proposed bisimulation loss.

In Figure 2, we show the state-action visitations of policies learned under each algorithm. As expected, the policy learned by filtered BC primarily takes the path towards the unsafe goal state. However, an optimal policy should take the path rightwards that turns the goal into the "safe" one. Both offline RL algorithms attempt to learn such a policy. However, the policy learned by naïve CQL sometimes fails to realize that it must take the rightward path from the start state in order to do so, resulting in a high proportion of failed trajectories. This is likely due to the policy failing to infer the correct goal state due to improperly discarding relevant information in its observation history (as RNNs are known to "forget" states that occur far in the past). As we hypothesized, adding a bisimulation loss remedied this issue, and the learned policy successfully takes the optimal path towards the "safe" goal state.

## 7.2 VISUAL NAVIGATION

Next, we consider a much more complex task with image observations. We aim to show that our proposed approach improves offline RL performance even when the observation space is large.

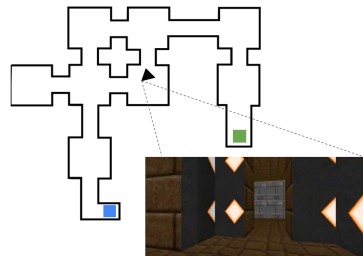

The task we consider involves navigating a maze from first-person pixel observations, namely the "My Way Home" scenario in the ViZDoom environment (Kempka et al., 2016). In the task, the agent starts in a random room (among 8 total rooms) at a random orientation, and is tasked to search for a piece of armor that is in a specific room. At each step, the agent observes a $320 \times 240$ rendering of its first-person view of the maze, which we cropped and resized to be $80 \times 80$ in our experiments. The agent has three available actions: $\{$turn left, turn right, and move forward$\}$. The figure on the left shows the layout and one possible observation by the agent. The reward at each state is $-0.0001$ except at the location of the armor, where it is $+1$, and the agent has $H = 2,100$ timesteps to find the armor. The starting location of the agent is unknown and must be inferred from history.

We construct a dataset of $|\mathcal{D}| = 5 \times 10^7$ frames, where $50\%$ of trajectories come from a policy that moves randomly, and $50\%$ from a policy trained via A2C (Mnih et al., 2016) on roughly $5 \times 10^6$ frames. The policy performs better than random, but still only successfully solves the task $60\%$ of the time. However, we posit that both the random and A2C policies will occasionally behave optimally on different subsets of the maze, trajectory stiching will enable the learning of a policy that drastically improves upon both of them. We consider the following baselines, all of which use the same CNN and RNN to process the observation histories: (1) behavioral cloning (BC) on the full dataset, (2) filtered BC on the $40\%$ of trajectories in the data with highest reward, (3) conservative Q-learning (CQL) (Kumar et al., 2020), (4) CQL augmented with our proposed bisimulation loss, (5) implicit Q-learning (IQL) (Kostrikov et al., 2021), (6) IQL with bisimulation loss, and (7) offline model-based policy optimization (MOPO) (Yu et al., 2020). CQL are IQL are both state-of-the-art offline algorithms; meanwhile, MOPO is a model-based offline algorithm where an ensemble of models is used to generate synthetic rollouts, whereas in our proposed approach, a model is also learned but solely used in a contrastive loss.

In Table 1, we show the cumulative rewards achieved by each algorithm across 100 independent evaluations. In the "base" task, the agent spawns in a random location, and in the "hard" task, the agent always spawns in the room farthest from the goal (blue in above figure). We see that offline RL greatly outperforms imitation learning in each environment, and that adding our bisimulation

loss noticeably improves performance. We also see that the improvement is greater in the "hard" task, likely because trajectories are longer and learning compact representations is more important. Finally, we observe that using a learned model in a bisimulation loss is much more effective than in traditional model-based optimization, which are likely more sensitive to model bias.

### 7.3 NATURAL LANGUAGE GAME

Our final task is a challenging benchmark to test the capabilities of offline RL on a natural language task. In particular, we aim to learn agents that successfully play the popular game Wordle. We adopt the details from this task from Snell et al. (2023), but provide a summary below. Although this is a relatively simple task, we use real transformer-based language models to address it, providing an initial evaluation of our hypothesis at a scale similar to modern deep networks.

In the game, the agent tries to guess a 5-letter word randomly selected from a vocabulary of 400 words. Here, the state is the word and is completely unknown to the agent, and actions consists of a sequence of 5 letter tokens. After each action, the agent observes a sequence of 5 color tokens, each with one of three "colors" for each letter in the guessed word: "black" means the guessed letter is not in the underlying word, "yellow" means the guessed letter is in the word but not in the right location, and "green" means the guessed letter is in the right location. We give a reward of -1 for each incorrect guess and a reward of 0 for a correct guess, at which point environment interaction ends. The agent gets a maximum of $H = 6$ guesses to figure out the word.

We use a dataset of Wordle games played by real humans and scraped from tweets, which was originally compiled and processed by Snell et al. (2023). We train four algorithms that use GPT-2 (with randomly initialized parameters) as a backbone transformer that encodes observation histories. The supervised methods predict actions via imitation learning as an additional head from the transformer: (1) fine-tuning (FT) uses the entire data, and (2) filtered FT uses top-25% of trajectories. The offline RL methods are: (3) Implicit Language Q-learning (ILQL) (Snell et al., 2023), and (4) ILQL with bisimulation loss.

| Method | Wordle Score |
|---|---|
| Fine-tuning | $-2.83 \pm 0.05$ |
| Filtered Fine-tuning | $-3.02 \pm 0.06$ |
| ILQL | $-2.21 \pm 0.03$ |
| ILQL + Bisimulation | $\mathbf{-2.19} \pm 0.03$ |

Table 2: Mean and standard deviation of scores achieved after training on human Wordle dataset.

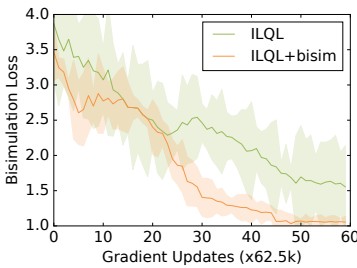

Figure 3: Bisimulation training loss.

We report mean and standard deviation of scores of all method across 200 independent evaluations in Table 2. We see that ILQL with bisimulation learning outperforms all other considered approaches, but only marginally over base ILQL. We hypothesize that the reason why base ILQL already performs well on the Wordle task is because standard training is already learning useful representations that induce a bisimulation metric. We assess whether this is true by measuring our bisimulation loss for ILQL both with and without explicit minimization of the loss in Figure 3 across 5 random runs of each algorithm. We notice that ILQL already implicitly minimizes the proposed loss during standard training. This is in line with our hypothesis, though is somewhat surprising, as ILQL has no awareness of yet still reduces the loss during training.

## 8 DISCUSSION

In this paper, we study the effectiveness of offline RL algorithms in POMDPs with unknown state spaces, where policies must utilize observation histories. We prove that because offline RL cannot in the worst case benefit from "trajectory stitching" to learn efficiently in POMDPs, it suffers from poor worst-case sample complexity. However, we also identify that offline RL can actually be provably efficient with suitable representations. Such representations discard features irrelevant for action selection. We show a sufficient condition for this when the representations induce a bisimulation metric. In addition, we show how to improve existing offline RL algorithms by adding a bisimulation loss to enforce the learning of such representations. While we show that learning representations that induce a bisimulation metric is sufficient to improve the effectiveness of offline RL with observation histories, it is by no means *necessary*. A direction for future work is deriving a more nuanced characterization of when useful representations are learned just by standard offline RL training.

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

# A PROOFS

In this section, we show the full proofs for the lemmas and theorems described in the main paper.

## A.1 PROOF OF THEOREM 4.1

In this section, we proof the performance guarantee for PEVI given by the pseudocode in Algorithm 2. We follow the proof of Theorem 4.2 in Kumar et al. (2022), but adapt it for our setting of observation-history-MDPs.

---

**Algorithm 2** PEVI

---

**Require:** Offline dataset $\mathcal{D}$, confidence level $\delta$
1: Compute $n(\tau, a)$ from $\mathcal{D}$, and estimate $\widehat{r}(\tau, a)$, $\widehat{P}(\cdot | \tau, a)$, $\forall (\tau, a) \in \mathcal{H} \times \mathcal{A}$
2: Initialize $\widehat{Q}(\tau, a) \leftarrow 0, \widehat{V}(\tau) \leftarrow 0, \forall (\tau, a)$
3: **for** $h = 1, 2, \ldots, H$ **do**
4:     Construct $c(\tau, a), \forall (\tau, a) \in \mathcal{H} \times \mathcal{A}$ based on $\mathcal{D}$.
5:     Set $\widehat{Q}(\tau, a) \leftarrow \widehat{r}(\tau, a) - c(\tau, a) + \sum_{o'} \widehat{P}(o' | \tau, a) \cdot \widehat{V}(\tau \oplus o'), \forall (\tau, a) \in \mathcal{H} \times \mathcal{A}$.
6:     Set $\widehat{\pi}(\cdot | \tau) \leftarrow \arg\max_{\pi} \widehat{Q}(\tau, \cdot) \cdot \pi(\cdot | \tau), \forall \tau \in \mathcal{H}$.
7:     Set $\widehat{V}(\tau) \leftarrow \widehat{Q}(\tau, \cdot) \cdot \widehat{\pi}(\cdot | \tau), \forall \tau \in \mathcal{H}$.
8: Return $\widehat{\pi}$

---

Recall from Section 4, that in a tabular POMDP, confidence intervals $c(\tau, a), \forall (\tau, a) \in \mathcal{H} \times \mathcal{A}$ can be constructed as in Equation 1.

### A.1.1 TECHNICAL LEMMAS

**Lemma A.1** (Bernstein's inequality). *Let $X, \{X_i\}_{i=1}^n$ be i.i.d random variables with values in $[0, 1]$, and let $\delta > 0$. Then we have*

$$\mathbb{P}\left(\left|\mathbb{E}[X] - \frac{1}{n}\sum_{i=1}^n X_i\right| > \sqrt{\frac{2\mathrm{Var}(X)\log(2/\delta)}{n}} + \frac{\log(2/\delta)}{n}\right) \leq \delta.$$

**Lemma A.2** (Theorem 4, Maurer and Pontil (2009)). *Let $X, \{X_i\}_{i=1}^n$ with $n \geq 2$ be i.i.d random variables with values in $[0, 1]$. Define $\bar{X} = \frac{1}{n}\sum_{i=1}^n X_i$ and $\widehat{\mathrm{Var}}(X) = \frac{1}{n}\sum_{i=1}^n(X_i - \bar{X})^2$. Let $\delta > 0$. Then we have*

$$\mathbb{P}\left(\left|\mathbb{E}[X] - \frac{1}{n}\sum_{i=1}^n X_i\right| > \sqrt{\frac{2\widehat{\mathrm{Var}}(\bar{X})\log(2/\delta)}{n-1}} + \frac{7\log(2/\delta)}{3(n-1)}\right) \leq \delta.$$

**Lemma A.3** (Lemma 4, Ren et al. (2021)). *Let $\lambda_1, \lambda_2 > 0$ be constants. Let $f : \mathbb{Z}_{\geq 0} \to \mathbb{R}$ be a function such that $f(i) \leq H, \forall i$ and $f(i)$ satisfies the recursion*

$$f(i) \leq \sqrt{\lambda_1 f(i+1)} + \lambda_1 + 2^{i+1}\lambda_2.$$

*Then, we have that $f(0) \leq 6(\lambda_1 + \lambda_2)$.*

### A.1.2 PESSIMISM GUARANTEE

The first thing we want to show is that with high probability, the algorithm provides pessimistic value estimates, namely that $\widehat{V}_h(\tau) \leq V^*(\tau)$ for all $h \in [H]$ and $\tau \in \mathcal{H}$. To do so, we introduce a notion of a "good" event, which occurs when our empirical estimates of the MDP are not far from the true MDP. We define $\mathcal{E}_1$ to be the event where

$$\left|(\widehat{P}(\cdot | \tau, a) - P(\cdot | \tau, a)) \cdot \widehat{V}_h(\tau \oplus \cdot)\right| \leq \sqrt{\frac{H\mathbb{V}(\widehat{P}(\cdot | \tau, a), \widehat{V}_h(\tau \oplus \cdot))\iota}{(n(\tau, a) \wedge 1)}} + \frac{\iota}{(n(\tau, a) \wedge 1)} \quad (3)$$

holds for all $i \in [m]$ and $(\tau, a) \in \mathcal{H} \times \mathcal{A}$. With abuse of notation, we let $\widehat{V}_h(\tau \oplus \cdot)$ be a vector of values of histories of the form $\tau \oplus o'$ for $o' \in \mathcal{O}$. We also define $\mathcal{E}_2$ to be the event where

$$|\widehat{r}(\tau, a) - r(\tau, a)| \le \sqrt{\frac{\widehat{r}(\tau, a)\iota}{(n(\tau, a) \wedge 1)}} + \frac{\iota}{(n(\tau, a) \wedge 1)} \tag{4}$$

holds for all $(\tau, a)$.

We want to show that the good event $\mathcal{E} = \mathcal{E}_1 \cap \mathcal{E}_2$ occurs with high probability. The proof mostly follows from Bernstein's inequality in Lemma A.1 . Note that because $\widehat{P}(\cdot \mid \tau, a), \widehat{V}_i$ are not independent, we cannot straightforwardly apply Bernstein's inequality. We instead use the approach of Agarwal et al. (2020) who, for each state $s$, partition the range of $\widehat{V}_i(\tau)$ within a modified $s$-absorbing MDP to create independence from $\widehat{P}$. The following lemma from Agarwal et al. (2020) is a result of such analysis:

**Lemma A.4** (Lemma 9, Agarwal et al. (2020)). *For any $h \in [H]$, $(\tau, a) \in \mathcal{H} \times \mathcal{A}$ such that $n(\tau, a) \ge 1$, and $\delta > 0$, we have*

$$\mathbb{P}\left( \left| (\widehat{P}(\cdot \mid \tau, a) - P(\cdot \mid \tau, a)) \cdot \widehat{V}_h(\tau \oplus \cdot) \right| > \sqrt{\frac{H\mathbb{V}(\widehat{P}(\cdot \mid \tau, a), \widehat{V}_h(\tau \oplus \cdot))\iota}{n(\tau, a)}} + \frac{\iota}{n(\tau, a)} \right) \le \delta \,.$$

Using this, we can show that $\mathcal{E}$ occurs with high probability:

**Lemma A.5.** $\mathbb{P}(\mathcal{E}) \ge 1 - 2|\mathcal{H}||\mathcal{A}|H\delta$.

*Proof.* For each $i$ and $(\tau, a)$, if $n(\tau, a) \le 1$, then equation 3 and equation 4 hold trivially. For $n(\tau, a) \ge 2$, we have from Lemma A.4 that

$$\mathbb{P}\left( \left| (\widehat{P}(\cdot \mid \tau, a) - P(\cdot \mid \tau, a)) \cdot \widehat{V}_h(\tau \oplus \cdot) \right| > \sqrt{\frac{H\mathbb{V}(\widehat{P}(\cdot \mid \tau, a), \widehat{V}_h(\tau \oplus \cdot))\iota}{n(\tau, a)}} + \frac{\iota}{n(\tau, a)} \right) \le \delta \,.$$

Similarly, we can use Lemma A.2 to derive

$$\mathbb{P}\left( |\widehat{r}(\tau, a) - r(\tau, a)| > \sqrt{\frac{\widehat{r}(\tau, a)\iota}{n(\tau, a)}} + \frac{\iota}{n(\tau, a)} \right)$$

$$\le \mathbb{P}\left( |\widehat{r}(\tau, a) - r(\tau, a)| > \sqrt{\frac{\widehat{\text{Var}}(\widehat{r}(\tau, a))\iota}{2(n(\tau, a) - 1)}} + \frac{\iota}{2(n(\tau, a) - 1)} \right) \le \delta \,,$$

where we use that $\widehat{\text{Var}}(\widehat{r}(\tau, a)) \le \widehat{r}(\tau, a)$ for $[0, 1]$ rewards, and with slight abuse of notation, let $\iota$ capture all constant factors. Taking the union bound over all $i$ and $(\tau, a)$ yields the desired result. $\square$

Now, we can prove that our value estimates are indeed pessimistic.

**Lemma A.6** (Pessimism Guarantee). *On event $\mathcal{E}$, we have that $\widehat{V}_h(\tau) \le V^{\widehat{\pi}}(\tau) \le V^*(\tau)$ for any step $h \in [H]$ and state $\tau \in \mathcal{H}$.*

*Proof.* We aim to prove the following for any $h$ and $\tau$: $\widehat{V}_{h-1}(\tau) \le \widehat{V}_h(\tau) \le V^{\widehat{\pi}}(\tau) \le V^*(\tau)$. We prove the claims one by one.

$\widehat{V}_{h-1}(\tau) \le \widehat{V}_h(\tau)$: This is directly implied by the monotonic update of our algorithm.

$\widehat{V}_h(\tau) \le V^{\widehat{\pi}}(\tau)$: We will prove this via induction. We have that this holds for $\widehat{V}_0$ trivially. Assume it holds for $h - 1$, then we have

$$V^{\widehat{\pi}}(\tau) \ge \mathbb{E}_{a \sim \widehat{\pi}(\cdot \mid \tau)} \left[ r(\tau, a) + P(\cdot \mid \tau, a) \cdot \widehat{V}_{h-1}(\tau \oplus \cdot) \right]$$

$$\ge \mathbb{E}_a \left[ \widehat{r}(\tau, a) - c_h(\tau, a) + \widehat{P}(\cdot \mid \tau, a) \cdot \widehat{V}_{h-1}(\tau \oplus \cdot) \right] +$$

$$\qquad \mathbb{E}_a \left[ c_h(\tau, a) - (\widehat{r}(s, a) - r(\tau, a)) - (\widehat{P}(\cdot \mid \tau, a) - P(\cdot \mid \tau, a)) \cdot \widehat{V}_{h-1}(\tau \oplus \cdot) \right]$$

$$\ge \widehat{V}_h(\tau) \,,$$

where we use that

$$c_h(\tau, a) \geq (\widehat{r}(s, a) - r(\tau, a)) + (\widehat{P}(\cdot \mid \tau, a) - P(\cdot \mid \tau, a)) \cdot \widehat{V}_{h-1}(\tau \oplus \cdot)$$

under event $\mathcal{E}$.

Finally, the claim of $V^{\widehat{\pi}}(\tau) \leq V^*(\tau)$ is trivial, which completes the proof of our pessimism guarantee. $\qquad\square$

### A.1.3 PERFORMANCE GUARANTEE

Now, we are ready to derive the performance guarantee from Theorem 4.1. We start with the following value difference lemma for pessimistic offline RL:

**Lemma A.7** (Theorem 4.2, Jin et al. (2021a)). *On event $\mathcal{E}$, at any step $h \in [H]$, we have*

$$J(\pi^*) - J(\widehat{\pi}) \leq 2 \sum_{h=1}^{H} \sum_{(\tau, a)} d_h^*(\tau, a) c_h(\tau, a) \,, \tag{5}$$

*where $d_h^*(\tau, a) = \mathbb{P}(\tau_h = \tau, a_h = a; \pi^*)$ for $\tau_h = (o_1, a_1, \ldots, o_h)$.*

*Proof.* The proof follows straightforwardly from Jin et al. (2021a) for standard MDPs by simply replacing states with observation histories. $\qquad\square$

Now, we are ready to bound the desired quantity $\mathsf{SubOpt}(\widehat{\pi}^*) = \mathbb{E}_{\mathcal{D}}[J(\pi^*) - J(\widehat{\pi})]$. We have

$$\mathbb{E}_{\mathcal{D}}[J(\pi^*) - J(\widehat{\pi}^*)] = \mathbb{E}_{\mathcal{D}}\left[\sum_{\tau} \rho_1(\tau)(V^*(\tau) - V^{\widehat{\pi}}(\tau))\right] \tag{6}$$

$$= \underbrace{\mathbb{E}_{\mathcal{D}}\left[\mathbb{I}\{\bar{\mathcal{E}}\} \sum_{\tau} \rho_1(\tau)(V^*(\tau) - V^{\widehat{\pi}}(\tau))\right]}_{:=\Delta_1}$$

$$+ \underbrace{\mathbb{E}_{\mathcal{D}}\left[\mathbb{I}\{\exists \tau \in \mathcal{H}, \ n(\tau, \pi^*(\tau)) = 0\} \sum_{\tau} \rho_1(\tau)(V^*(\tau) - V^{\widehat{\pi}}(\tau))\right]}_{:=\Delta_2}$$

$$+ \underbrace{\mathbb{E}_{\mathcal{D}}\left[\mathbb{I}\{\forall \tau \in \mathcal{H}, \ n(\tau, \pi^*(\tau)) > 0\}\mathbb{I}\{\mathcal{E}\} \sum_{\tau} \rho_1(\tau)(V^*(\tau) - V^{\widehat{\pi}}(\tau))\right]}_{:=\Delta_3} .$$

We bound each term individually. The first is bounded as $\Delta_1 \leq \mathbb{P}(\bar{\mathcal{E}}) \leq 2|\mathcal{H}||\mathcal{A}|H\delta \leq \frac{H\iota}{N}$ for choice of $\delta = \frac{1}{2|\mathcal{H}||\mathcal{A}|HN}$.

**Bound on $\Delta_2$.** For the second term, we have

$$\Delta_2 \leq \sum_{\tau} \rho_1(\tau)\mathbb{E}_{\mathcal{D}}[\mathbb{I}\{n(\tau, \pi^*(\tau)) = 0\}]$$

$$\leq H \sum_{\tau} d^*(\tau, \pi^*(\tau))\mathbb{E}_{\mathcal{D}}[\mathbb{I}\{n(\tau, \pi^*(\tau)) = 0\}]$$

$$\leq C^* H \sum_{\tau} \mu(\tau, \pi^*(\tau))(1 - \mu(\tau, \pi^*(\tau)))^N$$

$$\leq \frac{4C^*|\mathcal{O}|}{9N} \,,$$

where we use that $\rho_1(\tau) = d^*(\tau, \pi^*(\tau))$ as $\tau = o_1$, and that $\max_{p \in [0,1]} p(1 - p)^N \leq \frac{4}{9N}$.

**Bound on $\Delta_3$.** What remains is bounding the last term, which we know from Lemma A.7 is bounded by

$$\Delta_3 \leq 2\mathbb{E}_{\mathcal{D}}\left[\mathbb{I}\{\forall \tau \in \mathcal{H},\ n(\tau, \pi^*(\tau)) > 0\} \sum_{h=1}^{H} \sum_{(\tau,a)} d_h^*(\tau, a) c_h(\tau, a)\right],$$

Recall that $c_h(\tau, a)$ is given by

$$b_h(\tau, a) = \sqrt{\frac{H\mathbb{V}(\widehat{P}(\cdot \mid \tau, a), \widehat{V}_{h-1}(\tau \oplus \cdot)\iota}{n(\tau, a)}} + \sqrt{\frac{H\widehat{r}(\tau, a)\iota}{n(\tau, a)}} + \frac{H\iota}{n(\tau, a)}$$

We can bound the summation of each term separately. For the third term we have,

$$\mathbb{E}_{\mathcal{D}}\left[\sum_{h=1}^{H} \sum_{(\tau,a)} d_h^*(\tau, a)\,\frac{H\iota}{n(\tau, a)}\right] \leq \sum_{h=1}^{H} \sum_{(\tau,a)} d_h^*(\tau, a)\mathbb{E}_{\mathcal{D}}\left[\frac{H\iota}{n(\tau, a)}\right]$$

$$\leq \sum_{\tau} \sum_{h=1}^{H} d_h^*(\tau, \pi^*(\tau))\,\frac{H\iota}{N\mu(\tau, \pi^*(\tau))}$$

$$\leq \frac{H\iota}{N} \sum_{\tau} \left(\sum_{h=1}^{H} d_h^*(\tau, \pi^*(\tau))\right) \frac{H}{\mu(\tau, \pi_h^*(\tau))}$$

$$\leq \frac{C^*|\mathcal{H}|H^2\iota}{N}.$$

Here we use Jensen's inequality and that $\sum_{h=1}^{H} d_h^*(\tau, a) \leq C^*\mu(\tau, a)$ for any $(\tau, a)$. For the second term, we similarly have

$$\mathbb{E}_{\mathcal{D}}\left[\sum_{h=1}^{H} \sum_{(\tau,a)} d_h^*(\tau, a)\sqrt{H\frac{\widehat{r}(\tau, a)\iota}{n(\tau, a)}}\right]$$

$$\leq \mathbb{E}_{\mathcal{D}}\left[\sqrt{\sum_{h=1}^{H} \sum_{(\tau,a)} d_h^*(\tau, a)\frac{H\iota}{n(\tau, a)}}\right]\sqrt{\sum_{h=1}^{H} \sum_{(\tau,a)} d_h^*(\tau, a)\widehat{r}(\tau, a)}$$

$$\leq \sqrt{\frac{C^*|\mathcal{H}|H^3\iota}{N}},$$

where we use Cauchy-Schwarz. Finally, we consider the first term of $b_h(\tau, a)$

$$\mathbb{E}_{\mathcal{D}}\left[\sum_{h=1}^{H} \sum_{(\tau,a)} d_h^*(\tau, a)\sqrt{H\frac{\mathbb{V}(\widehat{P}(\cdot \mid \tau, a), \widehat{V}_{h-1}(\tau \oplus \cdot)\iota}{n(\tau, a)}}\right]$$

$$\leq \mathbb{E}_{\mathcal{D}}\left[\sqrt{\sum_{h=1}^{H} \sum_{(\tau,a)} d_h^*(\tau, a)\frac{H\iota}{n(\tau, a)}}\right]\sqrt{\sum_{h=1}^{H} \sum_{(\tau,a)} d_h^*(\tau, a)\mathbb{V}(\widehat{P}(\cdot \mid \tau, a), \widehat{V}_{h-1}(\tau \oplus \cdot))}$$

$$\leq \sqrt{\frac{C^*|\mathcal{H}|H^2\iota}{N}}\sqrt{\sum_{h=1}^{H} \sum_{(\tau,a)} d_h^*(\tau, a)\mathbb{V}(\widehat{P}(\cdot \mid \tau, a), \widehat{V}_{h-1}(\tau \oplus \cdot))}.$$

Similar to what was done in Zhang et al. (2020); Ren et al. (2021) for finite-horizon MDPs, we can bound this term using variance recursion for finite-horizon observation-history-MDPs. Define

$$f(i) := \sum_{h=1}^{H} \sum_{(\tau,a)} d_h^*(\tau, a)\mathbb{V}(\widehat{P}(\cdot \mid \tau, a), (\widehat{V}_{h-1}(\tau \oplus \cdot))^{2^i}). \tag{7}$$

Using Lemma 3 of Ren et al. (2021), we have the following recursion:

$$f(i) \leq \sqrt{\frac{C^*|\mathcal{H}|H^2\iota}{N}} f(i+1) + \frac{C^*|\mathcal{H}|H^2\iota}{N} + 2^{i+1}(\Phi+1),$$

where

$$\Phi := \sqrt{\frac{C^*|\mathcal{H}|H^2\iota}{N}} \sqrt{\sum_{h=1}^{H}\sum_{(\tau,a)} d_h^*(\tau,a)\mathbb{V}(\widehat{P}(\cdot \mid \tau,a), \widehat{V}_{h-1}(\tau\oplus\cdot)) + \frac{C^*|\mathcal{H}|H^2\iota}{N}} \qquad (8)$$

Using Lemma A.3, we can bound $f(0) = \mathcal{O}\left(\frac{C^*|\mathcal{H}|H\iota}{N} + \Phi + 1\right)$. Using that for constant $c$,

$$\begin{aligned}
\Phi &= \sqrt{\frac{C^*|\mathcal{H}|H^2\iota}{N} f(0)} + \frac{C^*|\mathcal{H}|H^2\iota}{N} \\
&\leq \sqrt{\frac{C^*|\mathcal{H}|H^2\iota}{N}\left(\frac{cC^*|\mathcal{H}|H^2\iota}{N} + c\Phi + c\right)} + \frac{C^*|\mathcal{H}|H^2\iota}{N} \\
&\leq \frac{c\Phi}{2} + \frac{2cC^*|\mathcal{H}|H^2\iota}{N} + \frac{c}{2}
\end{aligned}$$

we have that

$$\Phi \leq c + \frac{4cC^*|\mathcal{H}|H^2\iota}{N}.$$

Substituting this back into the inequality for $\Phi$ yields,

$$\Phi = \mathcal{O}\left(\sqrt{\frac{C^*|\mathcal{H}|H^2\iota}{N}} + \frac{C^*|\mathcal{H}|H^2\iota}{N}\right)$$

Finally, we can bound

$$\Delta_3 \leq \sqrt{\frac{C^*|\mathcal{H}|H^2\iota}{N}} + \frac{C^*|\mathcal{H}|H^2\iota}{N}.$$

Combining the bounds for the three terms yields the desired result.

## A.2   PROOF OF LEMMA 5.1

Recall that the *on-policy bisimulation metric* for policy $\pi$ on an observation-history-MDP is given by:
$$d^\pi(\tau,\tau') = |r^\pi(\tau) - r^\pi(\tau')| + W_1\left(P^\pi(\cdot \mid \tau), P^\pi(\cdot \mid \tau')\right), \qquad (9)$$

We use the following lemma that states that $d^\pi$ satisfies the following:

**Lemma A.8** (Theorem 3, Castro (2019)). *Given any two observation histories $\tau, \tau' \in \mathcal{H}$ in an observation-history-MDP, and policy $\pi$,*
$$|V^\pi(\tau) - V^\pi(\tau')| \leq d^\pi(\tau,\tau').$$

*Proof.* The proof follows straightforwardly from Castro (2019); Ferns et al. (2012) for standard MDPs by simply replacing states with observation histories. □

Furthermore, recall that we have a summarized MDP $(\mathcal{Z}, \mathcal{A}, P, r, \rho_1, H)$ where observation histories are clustered using aggregator $\Phi$. Let us define the reward function and transition probabilities for policy $\pi$ in the summarized-MDP as

$$r^\pi(\Phi(\tau)) = \frac{1}{\xi(\Phi(\tau))} \int_{\zeta\in\Phi(\tau)} r^\pi(\zeta)d\xi(\zeta),$$

$$P^\pi(\Phi(\tau') \mid \Phi(\tau)) = \frac{1}{\xi(\Phi(\tau))} \int_{\zeta\in\Phi(\tau)} P^\pi(\Phi(\tau') \mid \zeta)d\xi(\zeta),$$

where $\xi$ is a measure on $\mathcal{H}$.

We have,

$$
\begin{aligned}
|V^*(\tau) - V^*(\Phi(\tau))| &= \left| r^*(\tau) - r^*(\Phi(\tau)) + \int_{o'} P^*(o' \mid \tau)V^*(\tau \oplus o')do' - \int_{z'} P^*(z' \mid \Phi(\tau))V^*(z')dz \right| \\
&\leq \frac{1}{\xi(\Phi(\tau))} \int_{\zeta \in \Phi(\tau)} |r^\pi(\tau) - r^\pi(\zeta)| + \left| \int_{\tau'} P^\pi(\tau' \mid \tau) - P^\pi(\tau' \mid \zeta)V^\pi(\tau')d\tau' \right| d\xi(\zeta) \\
&\quad + \frac{H-1}{H} \sup_\tau |V^*(\tau) - V^*(\Phi(\tau))|
\end{aligned}
$$

Using Lemma A.2 and the dual formulation of the $W_1$ metric yields

$$
\begin{aligned}
&\leq \frac{1}{\xi(\Phi(\tau))} \int_{\zeta \in \Phi(\tau)} |r^\pi(\tau) - r^\pi(\zeta)| + W_1(P^\pi(\cdot \mid \tau), P^\pi(\cdot \mid \zeta))d\xi(\zeta) \\
&\quad + \frac{H-1}{H} \sup_\tau |V^*(\tau) - V^*(\Phi(\tau))| \\
&\leq \frac{1}{\xi(\Phi(\tau))} \int_{\zeta \in \Phi(\tau)} d^\pi(\tau, \zeta)d\xi(\zeta) + \frac{H-1}{H} \sup_\tau |V^*(\tau) - V^*(\Phi(\tau))| \\
&\leq \frac{1}{\xi(\Phi(\tau))} \int_{\zeta \in \Phi(\tau)} \widehat{d}_\phi(\tau, \zeta)d\xi(\zeta) + \left\| \widehat{d}_\phi - d^* \right\|_\infty + \frac{H-1}{H} \sup_\tau |V^*(\tau) - V^*(\Phi(\tau))| \\
&\leq 2\varepsilon + \left\| \widehat{d}_\phi - d^* \right\|_\infty + \frac{H-1}{H} \sup_\tau |V^*(\tau) - V^*(\Phi(\tau))|,
\end{aligned}
$$

Taking the suprenum of the LHS and solving yields the desired result.

### A.3   Proof of Theorem 5.1

The proof follows straightforwardly from noting that

$$
J(\pi^*) - J(\widehat{\pi}) = \mathbb{E}_{\rho_1} \left[ V^*(\tau). - V^{\widehat{\pi}}(\tau) \right] \leq \mathbb{E}_{\rho_1} \left[ V^*(\Phi(\tau)) - V^{\widehat{\pi}}(\Phi(\tau)) \right] + 2H \left( \varepsilon + \left\| \widehat{d}_\phi - d^* \right\|_\infty \right),
$$

where the inequality follows from applying Lemma 5.1.

Bounding the first term simply follows from using the same proof as in Section A.1, except where the space of observation histories $\mathcal{H}$ is now replaced by the space of summarizations $\mathcal{Z}$. This yields the desired result.

## B    EXPERIMENT DETAILS

In this section, we provide implementation details of each evaluated algorithm in each of our experimental domains.

### B.1    GRIDWORLD EXPERIMENTS

**Network architecture.**    We use a single-layer RNN with hidden dimension 128 to encode observation histories, which consist of all the previously visited states (encoded as one-hot vectors). The output of the RNN is fed through a single-layer MLP of hidden dimension 256, whose output is the representation used to generate the next action (as a softmax distribution), as well as train on using the bisimulation loss.

**Training details.**    We use the hyperparameters reported in Table 3.

| Hyperparameter | Setting |
|---|---|
| CQL $\alpha$ | 0.1 |
| Bisimulation $\eta$ | 0.05 |
| Discount factor | 0.99 |
| Batch size | 32 |
| Number of updates per iteration | 200 |
| Number of iterations | 100 |
| Optimizer | AdamW |
| Learning rate | 3e-5 |

Table 3: Hyperparameters used during training.

### B.2    VIZDOOM EXPERIMENTS

**Network architecture.**    We use the same convolutional architecture as in Justesen and Risi (2018). Specifically, we use a three-layer CNN with filter sizes of $[32, 64, 32]$ and strides $[4, 2, 1]$, which produces 32 feature maps of size $7 \times 7$. Then, the flattened input size is fed to a dense layer size of hidden size 512, then into a three-layer RNN with hidden dimension 512 to encode observation histories. Finally, the output of the RNN is fed through a single-layer MLP of hidden dimension 512, whose output is the representation used to generate the next action (as a softmax distribution), as well as train on using the bisimulation loss.

**Training details.**    We use the hyperparameters reported in Table 4.

| Hyperparameter | Setting |
|---|---|
| CQL $\alpha$ | 0.05 |
| Bisimulation $\eta$ | 0.02 |
| Discount factor | 0.99 |
| Batch size | 64 |
| Number of updates per iteration | 700 |
| Number of iterations | 200 |
| Optimizer | AdamW |
| Learning rate | 7e-4 |

Table 4: Hyperparameters used during training.

### B.3    WORDLE EXPERIMENTS

**Network architecture.**    We use the GPT-2 small transformer architecture, with the weights initialized randomly. One head of the transformer is used to generate the next action (as a softmax

distribution over the set of 26 characters). All heads are two-layer MLPs with hidden dimension twice that of the transformer's embedding dimension.

**Training details.**   We use the hyperparameters reported in Table 5. All algorithms were trained on a single V100 GPU until convergence, which took less than 3 days.

| Hyperparameter | Setting |
|---|---|
| ILQL $\tau$ | 0.8 |
| ILQL $\alpha$ | 0.001 |
| Bisimulation $\eta$ | 0.02 |
| Discount factor | 0.99 |
| Batch size | 1024 |
| Target network update $\alpha$ | 0.005 |
| Number of updates per iteration | 60 |
| Number of iterations | 60 |
| Optimizer | AdamW |
| Learning rate | 1e-4 |

Table 5: Hyperparameters used during training.

