# OpenReview forum: "Offline RL with Observation Histories: Analyzing and Improving Sample Complexity"
_ICLR.cc/2024/Conference — ICLR 2024 poster_

### Official Review · Reviewer_dUhS · 2023-10-30

**Soundness:** 3 good
**Presentation:** 3 good
**Contribution:** 2 fair
**Rating:** 5
**Confidence:** 2

**Summary:**

This paper aims to implement offline reinforcement learning through better trajectory stitching. The main challenge here is that the problems being looked at involve sequences of observations rather than clear states. Because these states aren't well-defined, it's not easy to link similar ones together, making the "stitching" hard. To address this, the authors present an approach to learn a compact representation of history comprising only features relevant to action selection. They adopt a bisimulation loss to do this and show that with this approach, offline reinforcement learning can be more efficient in terms of the worst-case sample complexity.

**Strengths:**

1. The paper is written in a very clear and understandable manner.

2. The concept of stitching together trajectories based on observations rather than states is quite applicable, especially since in many real-world situations, we don't have direct access to the game's state.

3. The proposed solution is strongly backed by thorough theoretical analysis.

**Weaknesses:**

1. This solution appears to be heavily dependent on the concept of trajectory stitching, which might limit its usefulness in other offline reinforcement learning (RL) scenarios.

2. I’m a bit uncertain about the novelty of training an offline RL model based on observation histories. It seems that in most offline RL research, what an agent can access is already observations (rather than states) from previous data collections.

**Questions:**

Can you explain how your solution might work with other offline RL methods that don't focus much on trajectory stitching (for example models with a pessimistic formulation mentioned in the related works)?

---

> ### Author Response · Authors · 2023-11-19
> **Response to Reviewer dUhS**
>
> Thank you for your review. We aim to address concerns that the reviewer raised regarding the usefulness of our proposed approach.
>
> **Reliance on trajectory stitching**
>
> We would like to point out that trajectory stitching is not some restrictive condition, but rather a general concept that explains the sample-efficiency of offline RL. Namely, it is commonly believed that trajectory stitching is a critical component as to why offline RL methods outperform BC methods on the same data, and has been demonstrated by prior empirical [1] and theoretical work [2]. By claiming that our approach performs trajectory stitching more effectively in POMDPs, we are not making any assumptions on the applicability of our approach, but instead are just saying that our method can achieve greater sample-efficiency. Note that all the approaches we evaluate do not explicitly perform any trajectory stitching operation, but rather operate on the general principle of pessimism.
>
> [1] https://arxiv.org/pdf/2004.07219.pdf
>
> [2] https://arxiv.org/pdf/2204.05618.pdf
>
>
> **Novelty of training an offline RL model based on observation histories**
>
> We do not claim that our novelty is in conditioning offline RL on observation histories, as we agree that has been studied in multiple prior works by naively replacing the policy with a sequence model that operates on histories rather than states. In our opinion, what we offer in this work is a novel perspective as to why naive offline RL succeeds in MDPs but sometimes struggles in POMDPs (via the lens of trajectory stitching), and theoretical and empirical evidence that bisimulation is a simple but effective method to create useful representations of histories.

---

> ### Author Response · Authors · 2023-11-22
> **Let us know if you have further questions**
>
> Let us know if you have further questions or concerns. We will do our best to answer them before the discussion period ends!

---

### Official Review · Reviewer_Hq7u · 2023-11-01

**Soundness:** 3 good
**Presentation:** 3 good
**Contribution:** 2 fair
**Rating:** 5
**Confidence:** 3

**Summary:**

The authors focus on the sample efficiency issue of  offline RL in the setting of POMDP with observation histories. Based on the observaton that two different trajectories would always have different histories, which makes stitching become much harder because no real state can be observed,  the authors propose to learn a more  realistic belief state using the bismulation loss and show that this may significantly reduce the sample complexity when working in this space because there is no need to consider complexity introduced by the observation sequence any more.

**Strengths:**

This paper show the usefulness of state abstraction in the setting of offline RL.

**Weaknesses:**

see below

**Questions:**

1) The reason that the bound of the sub-optimality of the learnt policy listed in Theorem 5.1 is much tighter than the bound listed in Theorem 4.1 is mainly due to the fact that the space sizes follow $|Z|<<|H|$, which is assumed. However, the state represenation space could still be very large. In addition, this bound does not work in non-tabular POMDPs.

2) The loss function $J(\phi)$ of the proposed method aims to clone the bisimulation metric $|\hat{s}(z,a)-\hat{r}(z',a')|-D(\hat{P}(\cdot|z,a),\hat{P}(\cdot|z',a'))$ . However, the target is approximately estimated via $\hat{r}$ and the dynamics $\hat{P}$, which in turn are estimated based on the offline dataset D. Hence in essential, the proposed method belongs to model-based offline RLs and should compare the proposed method to current model based offline RL methods.

3) The comparison study conducted in Table 1 may not be fair, because the three methods listed are not designed to learn from observation histories, but they can only get the information from the current sampled tuples. In this sense, the good  performance of the proposed method may due to its capturing additional information from the historical data through state abstraction.

4) We  suggest the authors  attach more experiments on some widely used standard benchmarks, such as MuJoCo suites, to evaluate the effectiveness and feasibility of the proposed methods.

---

> ### Author Response · Authors · 2023-11-19
> **Response to Reviewer Hq7u**
>
> Thank you for your review. You raised several important points that we address below. Per the review, we also include additional empirical results for an offline model-based algorithm (specifically MOPO) on the visual navigation task (see below for details).
>
> **State representation space could still be very large**
>
> We agree that the state representation space could still be large, but we believe for a vast majority of applications it will be much smaller than the observation history space. This is because the representation space can be viewed as the space of sufficient statistics of history, e.g. the location of the agent in navigation, or the underlying preferences/intent in dialogue.
>
> In addition, you correctly pointed out that our current theoretical analysis is only for tabular POMDPs. In the case of linear POMDPs, the analysis would proceed similarly, but instead of looking at the size of the representation space |Z| vs observation history space |H|, we instead look at their dimensionality. We will add to the Appendix an equivalent analysis for linear POMDPs demonstrating this result.
>
> **Comparison to model-based approaches**
>
> We agree that we do learn a model of the environment in order to implement our proposed bisimulation loss. However, we do not collect simulated rollouts using the model, which means our approach does not suffer as much as model-based approaches do when the model is biased. Existing offline model-based approaches also learn an ensemble of models in order to estimate uncertainty, which our approach does not require. Nevertheless, we add a comparison to MOPO [1], a state-of-the-art offline model-based approach in Table 1 of our updated paper. We see that MOPO does not reach the performance of our considered offline model-free approaches even without bisimulation. Though this gap may be alleviated with more systematic hyperparameter tuning (which we will do with more time), we feel that these preliminary results already demonstrate the advantage of our proposed approach over model-based ones.
>
> [1] https://arxiv.org/pdf/2005.13239.pdf
>
> **Comparison … may not be fair**
>
> We evaluate the current state-of-the-art approaches to learning in POMDPs, which adapt existing approaches by having them operate over full observation histories using a sequence model. Our method is not given any more information than these baselines, so we believe that the comparison is fair. It is true that our method may be performing better from “capturing additional information,” but we would argue that is intended as this shows that traditional approaches do not effectively summarize all the important details from histories.
>
> **Evaluation on MuJoCo**
>
> The reason why we do not evaluate on MuJoCo is because tasks in that benchmark are fully-observed MDPs. Though it is possible to engineer POMDPs by masking parts of the state, we thought that they would not reflect realistic instances of partial-information. However, we would be happy to evaluate our approach on other POMDP benchmarks that were not considered in our paper.

---

> ### Author Response · Authors · 2023-11-22
> **Let us know if you have further questions**
>
> Let us know if you have further questions or concerns, and we will be happy to address them before the discussion period ends!

---

### Official Review · Reviewer_HexG · 2023-11-01

**Soundness:** 3 good
**Presentation:** 3 good
**Contribution:** 3 good
**Rating:** 5
**Confidence:** 3

**Summary:**

The paper focuses on demonstrating that "trajectory stitching," a characteristic of offline RL algorithms in standard MDPs, is absent when these algorithms are applied in POMDPs, where observation histories influence the training and action selection of the policy. It begins by effectively motivating the issue with theoretical insights and subsequently proposes a bisimulation metric. This metric simplifies the observation histories, and this metric is readily usable by any offline RL algorithm.

**Strengths:**

1. The paper is well-motivated and is easy-to-follow.
2. It thoroughly explores the theory necessary to both - highlight the issues with current methods and introduce the proposed solution.
3. The introduced bisimulation metric is versatile, easily integrating with popular offline RL algorithms.

**Weaknesses:**

1. The paper, despite its detailed theoretical sections, lacks comprehensive empirical evidence to substantiate its claims. It’s unclear why the metric was exclusively tested and compared with CQL, omitting comparisons with more recent offline RL baselines such as IQL. Evaluation of this metric in a continuous action space environment, compared against IQL or XQL, would be beneficial.

2. The observed performances of CQL+Bisimulation and vanilla CQL in VizDoom are notably similar, as is the case with ILQL with and without bisimulation in Section 7.3. Does this imply that the bisimulation metric primarily enhances performance in less complex environments like GridWorld?

**Questions:**

1. Currently the results are presented solely in singleton games. Can the findings of this paper be extended to other environments, particularly where training and testing distributions vary?
2. What criteria were used to determine the percentage of trajectories to be omitted in Filtered BC?
3. In reference to Table 2, do the reported results encompass the entire human World dataset, or is it a selected subset? Additionally, how effectively does the method execute in the online Wordle environment? (see https://github.com/Sea-Snell/Implicit-Language-Q-Learning#playing-wordle)

---

> ### Author Response · Authors · 2023-11-19
> **Response to Reviewer HexG**
>
> Thank you for your review. Your primary concerns were with the empirical evaluations of our paper, which we aim to address by incorporating comparisons to additional baselines. We would also be happy to compare against other relevant POMDP tasks if appropriate ones come to mind. We view our empirical evaluations as supplementing our theoretical predictions, and believe both aspects are substantive contributions of our paper.
>
> **Lacks comprehensive empirical evidence**
>
> We arbitrarily chose CQL as one such offline RL algorithm that achieves state-of-the-art performance in our visual navigation tasks, and ILQL for our language task. We agree that a more comprehensive empirical study should include a comparison to more baselines. To remedy this, we consider the additional baselines IQL and IQL+bisimulation, and a model-based baseline MOPO [1]  to our visual navigation experiment on VizDoom. In Table 1 of our updated paper, we see that IQL and CQL achieve fairly similar performance, and that bisimulation improves performance by 10-20%. For our language task, we are not aware of an offline RL algorithm competitive with ILQL, but would be happy to compare against it if any comes to the reviewer's mind.
>
> [1] https://arxiv.org/pdf/2005.13239.pdf
>
> **Observed performances of CQL+Bisimulation and vanilla CQL are .. similar**
>
> We agree that ILQL+bisimulation and vanilla ILQL are similar in the language task (and we show why by plotting the bisimulation loss). However, we believe that the performance difference between CQL+bisimulation and vanilla CQL in the visual navigation task is significant. Specifically, CQL+bisimulation outperforms CQL by ~15% on the base task, and over 25% on the hard task, which in our opinion, are statistically significant improvements, especially with how our proposed approach is a simple modification on top of the base algorithm.
>
> **Answers to questions**
>
> We also will answer the questions that the reviewer raised below (we incorporated answers to these questions in the updated paper):
>
> 1. The capability of our approach to handle distribution shift comes from the base offline RL algorithm that takes actions pessimistically. We do not anticipate that our approach, which is aimed at improving sample-efficiency in POMDPs, will additionally assist in distribution shift in any way. Note that all our evaluation domains exhibit the distribution shift problem just like in any other offline RL benchmark.
>
> 2. The filtering percentage was a tunable parameter within the range of 20-50%. We chose the percentage that performed the best during evaluation.
>
> 3. The results are for a subset of the entire vocabulary, consisting of 400 words, that was considered in the original ILQL paper.

---

> > ### Comment · Reviewer_HexG · 2023-11-20
> > **Response from reviewer**
> >
> > Thank you for your additional experiments in the ViZDoom environment. While I appreciate your efforts, my primary concern about the lack of empirical evidence still holds. The results, as they currently stand, show only a marginal improvement with bisimulation, as indicated by the overlapping error bars. Consequently, this does not provide strong empirical evidence to alleviate my concerns. Therefore, I will maintain my initial evaluation score.

---

> ### Author Response · Authors · 2023-11-22
> **Regarding empirical results**
>
> Thank you for your prompt response. Regarding the VizDoom experiments, though the error bars overlap in the base task, we believe the order of improvement from adding a simple loss is still notable (on par with the improvement of offline RL algorithms in comparably difficult domains [1, 2]). In addition, the improvement is statistically significant (much larger than the error bars) in the hard task where histories are guaranteed to be long.
>
> Furthermore, we understand that the reviewer is unconvinced by the empirical results that bisimulation improves offline RL with POMDPs. However, we would emphasize that we do not claim that bisimulation improves the performance of offline RL across the board. The main contribution of our paper as stated in Section 1 is to analyze when and how offline RL can be made sample-efficient when forced to condition on histories. Our focus is on developing a theoretical understanding of when offline RL with histories does or does not work well. The theoretical analysis shows that it works well when the algorithm learns a state representation that approximately satisfies a bisimulation condition. Our experiments aim to evaluate whether this is true, and in this respect we believe the experiments are very successful: when the bisimulation loss is minimized (either explicitly or implicitly), the performance of the offline RL algorithm improves. However, it is not the case that explicitly enforcing this bisimulation loss is always necessary -- in some cases even a standard RL algorithm already decreases this loss, as in the case of the LLM experiment in Figure 3. This should not be surprising: our theory does not indicate that naive offline RL algorithms always fail, just that their worst-case performance is poor. We believe that our paper is quite explicit in avoiding the claim that the bisimulation loss is actually required for good performance, but we would be happy to clarify that if it is unclear. However, we would emphasize that, given the currently poor understanding of offline RL with POMDPs, it is much more valuable to provide a clearer understanding of the problem than to offer yet another representation learning, and we therefore do not view it as a weakness of the work that the bisimulation loss is not always necessary for good performance.
>
> [1] https://arxiv.org/pdf/2110.06169.pdf
>
> [2] https://arxiv.org/pdf/2212.04607.pdf

---

### Meta-Review · Area_Chair_d5Mi · 2023-12-04

**Metareview:**

The paper describes a new technique for partially observable offline RL.  The technique learns sufficient statistics by bisimulation to approximate the observation history.  This can improve the sample complexity.  This is nice work that introduces a theoretically principled approach that is demonstrated in tabular, visual and textual domains.  While the reviewers requested more comprehensive experiments, the main contribution is the theoretically principled approach and the experiments seem suitable to provide evidence that the approach works as expected.

**Justification For Why Not Higher Score:**

While the proposed approach is interesting and effective, it does not represent a breakthrough.

**Justification For Why Not Lower Score:**

While the reviews are lukewarm (unanimous border line rejection), they did not provide feedback about the algorithm and the theory behind it, which leads me to believe that the reviewers simply did not understand the paper.

---

### Decision · Program_Chairs · 2024-01-16

Accept (poster)